# Study on Spatial Structure Characteristics of the Tourism and Leisure Industry

**Mingyu Zhao and Jianguo Liu \***

Tourism College, Beijing Union University, Beijing 100101, China; zhaomingyumingyu@163.com
* Correspondence: liujianguo@buu.edu.cn

**Abstract:** By taking Beijing as the case site, using open-source Point of Interest data, and employing spatial visualization techniques, this study explores the spatial structural characteristics of the Beijing tourism and leisure industry and its sub-sectors. It has been found that (1) the nearest neighbor indexes of the tourism and leisure industry and its sub-sectors are all less than 1, indicating that the tourism and leisure industry and its sub-sectors in Beijing exhibit a spatial clustering distribution. Scenic spots have the largest R-value of 0.52 and, thus, the lowest degree of clustering. The minimum R-value of 0.15 is found in catering, marking the highest degree of clustering in the industry; (2) the main directional trend of the tourism and leisure industry and its sub-sectors in Beijing is the "northeast-southwest" direction, the south-north directional dispersion is dominant, and scenic spots demonstrate a more noticeable trend of spatial dispersion; (3) within the area from Sanlitun Street in the north to Panjiayuan Street in the south, and from Chaoyangmen Street in the west to Liulitun Street in the east, is situated the largest portion of cluster centers with the highest degree of clustering in Beijing's tourism and leisure industry. The contiguous high-density cluster center of catering starts from Sanlitun Street in the north to Jinsong Street in the south, and from Chaoyangmen Street in the west to Liulitun Street in the east. The cluster of shopping and entertainment shows a checkerboard pattern in the CZCF and NUDZ. The high-value cluster of accommodation occurs primarily around Sanlitun, Panjiayuan, and Qianmen; (4) the distribution of three grades of hot spot areas and non-significant areas of tourism and leisure, catering, accommodation, and shopping and entertainment in Beijing demonstrates a circular pattern that centers around the CZCF and expands outward in sequence. High-value hot spot streets for this area are dominated by Beixinqiao Street, Hepingli Street, Sanlitun Street, Heping Street, and Tuanjiehu Street; and the high-value cold spot streets of the area are chiefly in Fuzizhuang Township, Wangping Town, Miaofeng Mountain Town, and Tanzhesi Town.

**Keywords:** tourism and leisure; spatial structure; Beijing

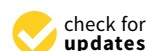

## 1. Introduction

Since China's reform and opening-up 40 years ago, its government has issued a series of policies and measures, such as the Opinions of the State Council on Accelerating the Development of Tourism, the Implementation Opinions on Encouraging and Guiding Private Capital Investment in Tourism, and the Outline of National Tourism and Leisure (2013–2020); accordingly, tourism and leisure activities have become an important way to improve the residents' life satisfaction [1]. Simultaneously, promoting high-quality development of the tourism and leisure industry acts as an effective driving force for the development of urban industries and China's transformation to a service-oriented economy. The spatial structure of the tourism and leisure industry represents the spatial distribution, spatial organization, and different spatial patterns of the economic activities of catering, accommodation, shopping, and scenic tours within the city. A disordered spatial structure of the tourism and leisure industry may cause urban problems in terms of traffic congestion, environmental degeneration, and higher management costs. In this context,

it is key to promote sustainable development of urban tourism. The reasonable spatial distribution of the tourism and leisure industry can effectively improve the efficiency of the tourism and leisure industry and reduce unnecessary waste, thus promoting the sustainable development of the tourism and leisure industry.

Several studies have observed noteworthy results while analyzing the various types of sub-sectors of the urban tourism and leisure industry. Overall, the spatial distribution of the urban catering industry shows polycentric characteristics with the degree of clustering varying along traffic arteries; usually with the high degree of clustering located in the city's core area and the low of that located in the peripheral areas [2–4]. On a large scale, the catering industry is sited around residential quarters, major business districts, and tourist attractions [5]; on a small scale, it is more distributed in public transport stops than residential areas [6]. Factors affecting the spatial distribution of different types of catering businesses differ, such that take-out O2O restaurants often rely on hospitals and universities [7], and fast-food restaurants are associated with residents' daily behavior patterns [8]. In addition, studies have found a correlation between the distribution patterns of restaurants in residential areas and obesity [9,10]. Studies on the accommodation industry include a description of its spatial and temporal evolution characteristics [11], factors influencing the spatial layout [12], and the spatial distribution pattern of prices and their influencing factors [13,14]. The research methods employed in the analysis of such influencing factors are mostly geographically weighted regression models, but as different scholars have studied different types of accommodation businesses, consistent conclusions are limited. The distribution of the shopping and entertainment industry in cities shows a typical "center-periphery" structure, chiefly shaped by the distribution of population and road networks [15–18]. With the development of mobile network technology and the logistics industry, the location choice of the new online-offline hybrid and demand data-oriented shopping industry is different from that of the traditional industry; the layout of the new industry is more decentralized and homogeneous, with a lower degree of clustering and a clear tendency toward polycentricity than that of the traditional industry [19,20]. In the future, new business formats and e-commerce will further change the spatial distribution pattern of urban land use and the shopping and entertainment industry [21]. Most scholars have studied the spatial structure of scenic spots at the provincial or city-group level and chosen A-class scenic spots as the research object, showing that high-density areas of scenic spots invariably appear around the provincial capital [22–26].

Previous studies on the identification of urban industry spatial structure are mostly based on conventional socio-economic statistics, census, questionnaire and interview data, and urban land use maps; research methods mainly include the rank-size rule, Pareto index, and Herfindahl index [27–29]. However, due to the heavy influence of human factors, such data exhibit weaknesses both in timeliness and accuracy and are often unable to capture the current status of urban industrial spatial structure. A large amount of geographic big data pertaining to people's daily lives have been generated with the development of information technology and mobile terminal devices; this has provided new data sources, such as data on social media check-ins, mobile phone signaling, nighttime lighting, and Point of Interest (POI), for the in-depth study of the urban industrial spatial structure. Among them, the open-source POI data are real-world point geographic entity data with the merits of easy access, large data volume, high accuracy, and fast updates [30]. POI data can reveal inner-city characteristics and reflect the spatial distribution of specific businesses in the city in more detail. To date, POI data have been mostly used for identifying urban polycentric spatial structure [31], the analysis of residents' daily activity characteristics [32–34] and personalized tourism route recommendations [35,36]; there are relatively few studies that have used POI data to reflect the overall structure of the urban tourism and leisure industry and its sub-sectors.

In recent years, the National Development and Reform Commission has proposed the policy of the decentralization of non-capital functions in Beijing. Following a steady

promotion of this policy, industries have been transferred to the surrounding areas in an orderly manner, such that the positive effects of reshaping the spatial pattern of various types of industries in the city have gradually emerged. Previous studies have focused on issues, such as the evolutionary characteristics of the cluster of productive service industries in Beijing [27], the spatial pattern of public service industries [37–41], and the spatial distribution characteristics of the catering and accommodation industries [2]. However, there are relatively few municipal-level studies analyzing the structure of the tourism and leisure industry in general while specifically exploring the spatial characteristics of its sub-sectors. Therefore, this study uses POI data and GIS spatial visualization technology to comprehensively describe the spatial layout characteristics of Beijing's tourism and leisure industry and its sub-sectors from three aspects—spatial distribution pattern, spatial cluster characteristics, and the distribution of hot and cold spots—in an attempt to get an overall picture of the spatial differentiation characteristics of the industry. Further, this can help in providing scientific reference for the optimization of its spatial layout planning.

This study uses Beijing as the research case, which is located in the northern part of the North China Plain. It has 16 districts including Chaoyang, Haidian, Dongcheng, and Xicheng, and the total area is 16,410.54 square kilometers. Beijing is the political, economic, and cultural center of China, and its level of economic development has been in the forefront of the country for a long time. Beijing is not only a modern international metropolis, but also a well-known historical capital at home and abroad. It has a splendid culture, beautiful natural scenery and various types of tourism resources, such as architectural monuments, leisure and entertainment places, and natural landscapes. Beijing also owns a well-developed transportation network and is one of China's railway network centers; this includes the Beijing-Kowloon Railway, Beijing-Guangzhou Railway, Beijing-Shanghai Railway, Beijing-Baotou Railway, etc. The Capital International Airport is one of the world's largest airports, with a tourist throughput of 10,013,600 in 2019. The number of passengers ranks first among domestic airports. The city's transportation is also very convenient, with subway and bus lines crisscrossed and dotted. Therefore, choosing Beijing as the case site is highly representative and typical, and exploring the spatial distribution characteristics of its tourism and leisure industry can provide a certain scientific reference for Beijing's tourism enterprise site selection, tourism route planning, and infrastructure layout.

## 2. Research Methodology and Data Sources

### 2.1. Research Methodology

#### 2.1.1. Average Nearest Neighbor Index

All POI data are related to point geographic entities. Of the various methods to analyze the distribution of point elements, the nearest neighbor index can more accurately and objectively determine the spatial distribution of point elements (cluster, randomness, or dispersion) and obtain the degree of agglomeration based on the data results [1]. The formula is as follows:

$$R = \frac{r_i}{r_E} = \frac{1}{n}\sum_{i=1}^{n} r_i(S_i) / \left(\frac{1}{2\sqrt{\frac{n}{A}}}\right) R = 1 \tag{1}$$

where $R$ is the nearest neighbor index; $r_E$ is the theoretical nearest neighbor distance under the random distribution of the elements; $n$ is the number of points; $r_i(S_i)$ is the distance from the point element to its nearest neighbor; and $A$ is the area of the district under study. When $R > 1$, the elements in the area are dispersed; $R < 1$ implies clustering, and $R = 1$ implies a random distribution. The smaller the $R$ value, the higher the degree of clustering [4].

### 2.1.2. Kernel Density Estimation

The kernel density estimation can clearly and visually represent the distribution density of point elements within an area, which in turn can be analyzed to ascertain the degree of clustering and the cluster center [6]. The formula is as follows:

$$f_n(x) = \frac{1}{nh^2} \sum_{\pi^i=1}^{n} K((1 - \frac{(x-x_i)^2 + (y-y_i)^2}{h^2}))^2 \tag{2}$$

where $f_n(x)$ is the value estimated from the sample; $h$ is the bandwidth; $n$ is the number of points whose distance from the point $(x,y)$ is less than $h$; $K$ is the kernel function; and $(x - x_i)^2 + (y - y_i)^2$ denotes the distance between the point $(x_i,y_i)$ and point $(x,y)$.

### 2.1.3. Standard Deviational Ellipse

The standard deviational ellipse (SDE) can be used to portray the outline of the data distribution and demonstrate the directional trend, dispersive trend, and central tendency of the spatial distribution of point data. Azimuth, major axis, and minor axis are important indicators of the SDE. Among them, the azimuth represents the main directional trend of the data distribution; the major axis indicates the dispersion of the data in the direction of the trend and of maximum diffusion; the minor axis represents the direction of minimum diffusion; the flatness is the ratio of the major axis to minor axis, with a higher flatness indicating a more obvious directionality; and the area illustrates the degree of data dispersion [15].

### 2.1.4. Spatial Autocorrelation Index

The global spatial autocorrelation index can be used to describe the overall spatial distribution of certain data attributes and ascertain the type of the spatial autocorrelation (e.g., high value cluster or low value cluster) by the attribute values [15]. This study adopts the most commonly used test statistic, Global Moran's I index, which is computed as:

$$Global\ Moran's\ I = \frac{\sum\limits_{i=1}^{n}\sum\limits_{j=1}^{n} w_{ij}(x_i - \overline{X})(x_j - \overline{X})}{s^2 \sum\limits_{i=1}^{n}\sum\limits_{j=1}^{n} w_{ij}} \tag{3}$$

where $s^2 = \frac{1}{n}\sum_{i=1}^{n}(x_i - \overline{x})$; $n$ is the number of spatial units; $x_i$, $x_j$ are the observations on spatial units $i$, $j$, respectively; and $w_{ij}$ is the spatial weight matrix. When Global Moran's I > 0, the cluster of spatial units tends to be high-high or low-low; Moran's I < 0, indicates a high-low cluster; and when Global Moran's I = 0, it indicates that there is no spatial autocorrelation. Hence, the spatial units are randomly distributed.

Based on the global spatial autocorrelation, the Getis-Ord $G_i^*$ test statistic was introduced for hot spot analysis, which describes the correlation of a geographical feature in a spatial unit with the same feature in the surrounding units. By comparing the local sum of the feature values for the unit in question and its adjacent units within a given distance with the global sum of all feature values, the statistic indicating the clustering degree of the feature at the local level can be calculated. High-value clustering areas are detected as hot spots, while low-value clustering areas are viewed as cold spots [19]. The formula is as follows:

$$G_i^* = \frac{\sum_{j=1}^{n} W_{ij}(d)X_j}{\sum_{j=1}^{n} X_j}(j \neq i) \tag{4}$$

where $W_{ij}$ is the contiguity-based spatial weights matrix. If the distance between the $i$-th and $j$-th spatial units is less than the critical distance d, then the element of the spatial

weight matrix is set to 1, otherwise 0. $X_j$ is the feature value of the $j$-th spatial unit. $n$ is the total number of units. $G_i^*$ is further normalized to

$$Z(G_i^*) = \frac{G_i^* - E(G_i^*)}{\sqrt{Var(G_i^*)}} \tag{5}$$

where $E(G_i^*)$ is the mathematical expectation, and $Var(G_i^*)$ is the coefficient of variation. When $Z(G_i^*)$ is positively significant, the area is a hot spot, indicating that values around spatial unit $i$ are high; when $Z(G_i^*)$ is negatively significant, it is a cold spot area, which indicates that the feature values around spatial unit i are low [7].

### 2.2. Data Sources

The POI data used in this study were collected from the June 2020 AMAP map (a digital map provided by AutoNavi based in China). As a virtual abstraction of spatial entities, POI data contain a series of attribute information of real geographic entities, such as the longitude, latitude, address, and category, that cover practically all physical objects in the study area, and are widely used in geographical research. In a general sense, tourism involves six aspects: food, accommodation, traveling, touring, shopping, and entertainment. Scenic spots are the core products of tourism, while food, shopping, and entertainment are primarily for the daily leisure services of city residents. To reflect the overall spatial structure characteristics of Beijing's tourism industry, this study incorporates sectors linked to tourism, such as food, accommodation, traveling, shopping, and entertainment, in the analysis. According to the classification strategy of AMAP and the purpose of this study, the POI data were divided into four categories: catering, accommodation, scenic spots, as well as shopping and entertainment, collectively referred to as tourism and leisure. The original data were screened, merged, with duplicate and misclassified data eliminated, to obtain a total of 362,249 outlet data as shown in Table 1.

**Table 1.** Beijing POI data classification.

| POI | Classification | Number/Pcs | Percentage/% |
|---|---|---|---|
| Catering | Chinese food, western food, fast food, beverage shop | 105,059 | 29.00 |
| Accommodation | Star-rated hotels, homestays, youth hostels | 28,983 | 8.00 |
| Scenic spots | A-level scenic spots, parks, squares | 4792 | 1.32 |
| Shopping and entertainment | Retail industry, integrated shopping center | 223,415 | 61.67 |
| Tourism and leisure | | 362,249 | 100 |

## 3. Analysis of Results

### 3.1. Spatial Distribution Patterns

Figure 1 illustrates the spatial distribution of the POIs of the tourism and leisure industry in Beijing as a circle, with a large number of POIs concentrated in the Core Zone of the Capital Function Zone (CZCF) (Dongcheng and Xicheng districts) and the Urban Function Extended Zone (UFEZ) (Chaoyang, Haidian, Fengtai, and Shijingshan districts), and fewer extending outward to the New Urban Development Zone (NUDZ) (Tongzhou, Shunyi, Daxing, Changping, and Fangshan districts) and the Eco-Conservation Zone (ECZ) (Mentougou, Pinggu, Huairou, Miyun, and Yanqing districts). The CZCF and the UFEZ are Beijing's major urban areas, with not just a long history, rich cultural heritage, and historical monuments, but a dense population, high rate of urbanization, high per capita income and consumption, and a developed tertiary industry. The NUDZ surrounding the major urban area is the main undertaker for nonessential functions of Beijing as the capital city. The broader ECZ is less densely populated, chiefly plays the role of an ecological

safety barrier with environmental protection as the top priority; therefore, it has a smaller amount of POIs in the tourism and leisure industry.

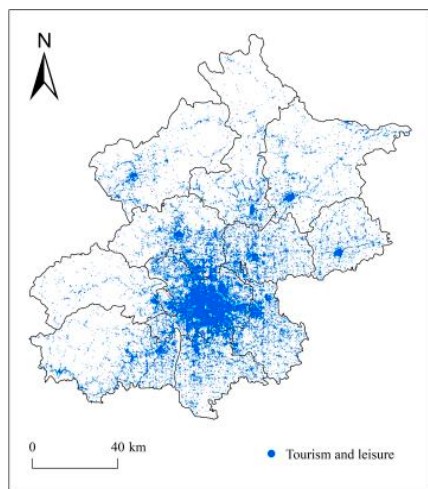

**Figure 1.** The spatial distribution of POI in Beijing's tourism and leisure industry.

Table 2 reveals that the nearest neighbor indexes are all less than 1, with a confidence level of 99%, indicating that the tourism and leisure industry and its sub-sectors in Beijing exhibit a spatial clustering distribution. The R-value suggests that the degrees of clustering vary to some degree among different sub-sectors, being ranked from high to low across catering, shopping and entertainment, accommodation, and scenic spots. Among them, scenic spots have the largest R-value of 0.52 and, thus, the lowest degree of clustering, which may be because the scenic spots in this study encompass a wide range of contents and varieties; they comprise not only conventional scenic spots but also various types of parks and squares, which are geographically more dispersed. In addition, the peri-urban NUDZ and ECZ have, in recent years, built various types of tourist attractions with the support of their natural resources, which to certain extent tones down the clustering of scenic spots. The minimum R-value of 0.15 is found in catering, marking the highest degree of clustering in the industry. As mentioned earlier, the catering industry tends to be concentrated around residential quarters or public transport stops [6]; hence, the clustering trend is more obvious.

**Table 2.** Tourism and leisure industry average nearest neighbor index.

|  | Tourism and Leisure | Catering | Accommodation | Scenic Spots | Shopping and Entertainment |
|---|---|---|---|---|---|
| R value | 0.18 | 0.15 | 0.21 | 0.52 | 0.19 |
| *p* value | 0.00 | 0.00 | 0.00 | 0.00 | 0.00 |

*3.2. Spatial Cluster Characteristics*

The nearest neighbor index clarifies whether the distribution pattern of point elements is clustered, random, or dispersive, but does not indicate the spatial orientation of the industry, nor the specific location of the cluster. Combining the SDE and kernel density to describe the spatial cluster characteristics of the tourism and leisure industry in Beijing, Figure 1 and Table 3 show that (1) the main directional trend of the tourism and leisure industry and its sub-sectors in Beijing is the "northeast-southwest" direction and that the south-north directional dispersion is dominant; (2) the length of the major axis reflects the degree of south-north directional dispersion of the industry, which declines in the order of scenic spots, accommodation, shopping and entertainment, and catering; (3) the minor axis reflects the degree of dispersion in the east-west direction, ranking from high

to low in scenic spots, accommodation, shopping and entertainment, and catering; (4) the most obvious directional sub-sector is accommodation, followed by scenic spots, shopping and entertainment, and catering; (5) the sub-sector with the widest diffusion is scenic spots, followed by accommodation, shopping and entertainment, and catering, which is consistent with the results of the nearest neighbor index analysis in Section 3.1, where scenic spots demonstrate a more noticeable trend of spatial dispersion.

**Table 3.** Tourism and leisure industry standard deviational ellipse.

|  | Tourism and Leisure | Catering | Accommodation | Scenic Spots | Shopping and Entertainment |
|---|---|---|---|---|---|
| Azimuth (°) | 39.10 | 36.07 | 36.77 | 36.07 | 40.99 |
| Major axis (km) | 32.53 | 28.92 | 40.17 | 48.44 | 32.55 |
| Minor axis (km) | 22.59 | 21.47 | 24.28 | 32.16 | 22.48 |
| Flatness | 1.44 | 1.35 | 1.65 | 1.50 | 1.44 |
| Area (km²) | 2308.27 | 1950.57 | 3063.79 | 4893.28 | 2298.87 |

The results of Figure 2 show that (1) Beijing's tourism and leisure industry has formed several cluster centers, with a gradual decrease in the degree of clustering outward from the CZCF. The central city is crowded, with contiguous and dense clustering areas, while distant suburbs are dotted with island-like clustering centers. Within the area from Sanlitun Street in the north to Panjiayuan Street in the south, and from Chaoyangmen Street in the west to Liulitun Street in the east, is situated the largest portion of cluster centers with the highest degree of clustering in Beijing's tourism and leisure industry, located at the junction of Dongcheng District and Chaoyang District, with most of the cluster centers in the latter. Around this piece-like clustering area, points of high-value secondary cluster centers are densely distributed in the shape of a semi-ring, with the west part near Chongwenmen in Dongcheng District and Financial Street and Guanganmen in Xicheng District, the north part near Zhongguancun in Haidian District and Wangjing in Chaoyang District, and the south part near Dahongmen in Fengtai District. The outer districts, including Fangshan, Changping, Yanqing, Huairou, Shunyi, and Miyun, each presenting a single isolated area with a small scale and high value of kernel density. Specifically, they are around Xilu Street in Fangshan District, Chengbei Street in Changping District, Yanqing Street in Yanqing District, Longshan Street in Huairou District, Shengli Street in Shunyi District, Gulou Street in Miyun District, and Binhe Street in Pinggu District; (2) catering and shopping and entertainment, which are closely related to the daily life of urban residents, have numerous widely-distributed POIs with similar spatial cluster characteristics as those shown by the tourism and leisure industry. The differences are mainly in the degree and scope of clustering. In the major urban area, the contiguous high-density cluster center of catering starts from Sanlitun Street in the north to Jinsong Street in the south, and from Chaoyangmen Street in the west to Liulitun Street in the east. The north-south scope of catering is smaller than that of the tourism and leisure industry. Beyond this core cluster center, two sub-centers are located to the west and north: in the west are Donghuamen Street in Dongcheng District and Financial Street in Xicheng District, while in the north are Zhongguancun in Haidian District and Wangjing in Chaoyang District. The number of secondary cluster centers for catering is smaller than that for tourism and leisure, as is the scope of clustering, but they coincide in specific locations of clustering and overlap to some degree in the cluster area. In Fangshan District, Changping District, and Shunyi District on the outskirts, there are three sporadic cluster centers for the catering industry, consistent with those of tourism and leisure, namely in Xilu Street, Chengbei Street, and Shengli Street; (3) the cluster of shopping and entertainment shows a checkerboard pattern in the CZCF and NUDZ without a large scale contiguous clustering center; instead, it exhibits multiple centers with comparable scopes and proximity. Specifically, they are near Sanlitun and Panjiayuan in Chaoyang District, Qianmen in Dongcheng District, Financial Street in Xicheng District, and Zhongguancun in Haidian District; (4) the high-value cluster

of accommodation occurs primarily around Sanlitun, Panjiayuan, and Qianmen, which are interconnected in the shape of a triangle. Secondary cluster centers are formed to its west near Anzhen Street, Beijing North Railway Station, Beijing West Railway Station, Zhongguancun, and Yongding Road, and to its north near Wangjing. Few island-like accommodation cluster centers are present in the ECZ, and there are two notable cluster centers only in Gubeikou Town in Miyun District and Shidu Town in Fangshan District; (5) scenic spots formulate only one large rectangular cluster center in the CZCF, which starts from Andingmen Street in the north to Dashilar Street in the south, and from Financial Street in the west to Jingshan Street in the east. The degree of clustering gradually decreases away from this center, displaying a typical circular distribution. A cluster center of scenic spots with high kernel density is not visible in the NUDZ or ECZ.

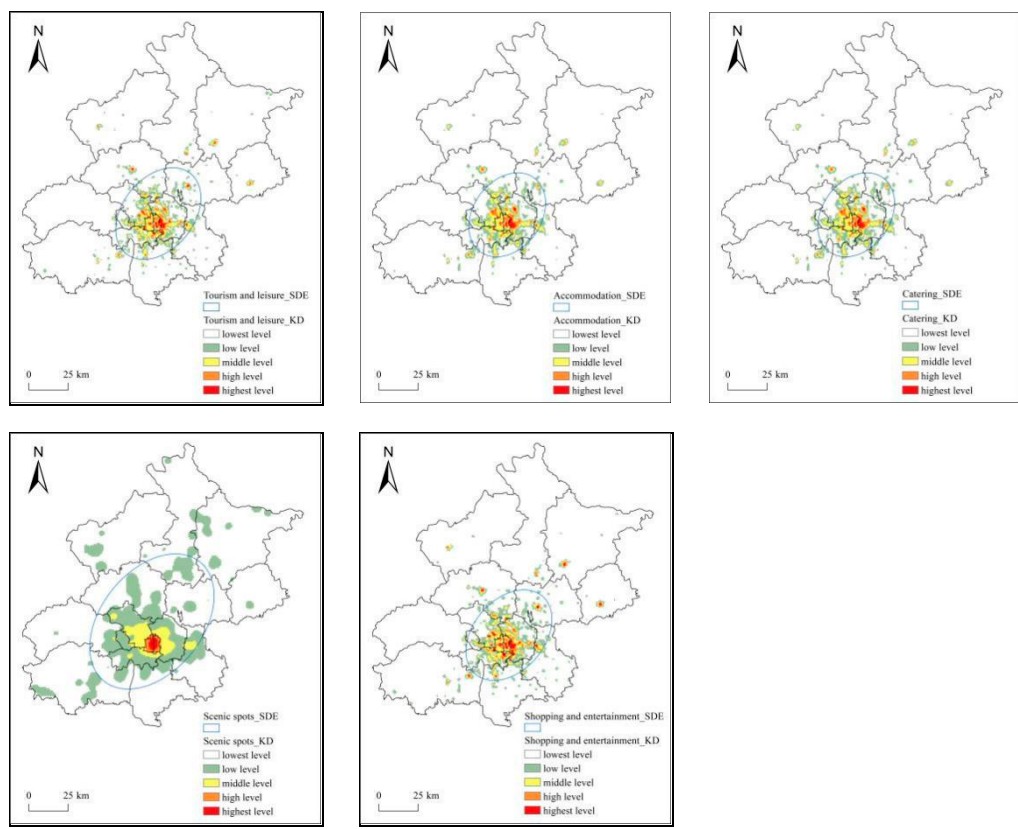

**Figure 2.** Tourism and leisure SDE and Standard KD.

### 3.3. Spatial Hot Spot Detection

The global Moran's index describes the overall autocorrelation of certain phenomena in space, and the magnitude of its attribute values determines whether the cluster is high-value with high value, low-value with low value, or high-value with low value. The results of the study show that the global Moran's index of Beijing's tourism and leisure industry, as well as its sub-sectors, is greater than 1 with a confidence level of 99%, indicating that there is a cluster of high-high and low-low types (Table 4). Based on this, the Getis-Ord $G_i^*$ test statistic was further applied, with the street as the unit of analysis, to identify the location of hot and cold spots in specific streets of Beijing. A street can be discerned as a hot spot if it has a high-density of outlets and is surrounded by other streets also with high densities of outlets; otherwise, it will be labeled a cold spot. Here, hot and cold spots are graded according to $Z(G_i^*)$ values as follows: a Grade I hot spot ($Z > 2.58$, $p < 0.01$), which denotes a highly significant hot spot area; a Grade II hot spot ($1.96 < Z < 2.58$, $p < 0.05$), which is a hot spot area of medium significance; a Grade III hot spot ($1.65 < Z < 1.96$, $p < 0.10$), which is a hot spot area of low significance; a Grade I cold spot ($Z < -2.58$, $p < 0.01$) or a highly

significant cold spot; a Grade II cold spot ($-2.58 < Z < -1.96$, $p < 0.05$) or a cold spot area of medium significance; a Grade III cold spot ($-1.96 < Z < -1.65$, $p < 0.10$) or a cold spot area of low significance; and finally, a random distribution area ($-1.65 < Z < 1.65$, $p > 0.10$), that is, an area without a significant cluster.

**Table 4.** Tourism and leisure industry Moran's I.

|  | Tourism and Leisure | Catering | Accommodation | Scenic Spots | Shopping and Entertainment |
|---|---|---|---|---|---|
| Global Moran's I | 0.19 | 0.24 | 0.38 | 0.08 | 0.14 |
| Z value | 16.55 | 20.28 | 32.74 | 6.94 | 11.91 |
| *p* value | 0.00 | 0.00 | 0.00 | 0.00 | 0.00 |

The results in Figure 3 and Tables 5 and 6 show that (1) the distribution of the three grades of hot spots and non-significant areas constitutes a circular pattern in Beijing's tourism and leisure industry, expanding outward layer by layer from the CZCF. Most of the hot spots are distributed in the CZCF and the UFEZ, a few in the NUDZ, and none in the ECZ. Of these, the Grade I hot spot area has the most streets in its scope, followed in turn by Grade II and Grade III hot spot areas. Respectively, they cover an area of 1722.22 km², 394.47 km², and 126.04 km² and contain 249,477, 12,393, and 5883 POIs. The cold spot areas are mainly located in the ECZ and the Shunyi and Fangshan Districts of the NUDZ. The first, second, and third-grade cold spot areas have an area of 1151.30 km², 2371.79 km², and 2743.85 km², respectively, and contain 4823, 14,256, and 8372 POIs, respectively. With regard to the areas of hot and cold spots and the number of POIs contained, the area of cold streets is 3 times larger than that of hot streets, but the number of POIs contained in hot streets is 10 times higher than that in cold streets. Tourism and leisure businesses in hot streets are more concentrated; (2) the spatial pattern of hot and cold spots for catering, entertainment, and shopping is the same as that for tourism and leisure, with most of the hot spots distributed in the CZCF and the UFEZ and a few located in the NUDZ, while the cold spots are mainly located in the ECZ and the Shunyi and Fangshan districts of the NUDZ. The number of blocks contained in the first, second, and third hot spots for catering is 138, 10, and 5, respectively, with areas of 1788.35 km², 196.03 km², and 296.38 km², and POIs of 249,477, 12,393, 5883, respectively. The number of streets contained in the first, second, and third-grade of hot spots in the entertainment and shopping sector is 134, 14, and 5, respectively, with an area of 1583.46 km², 549.78 km², and 107.48 km², and a quantity of POIs of 146,009, 12,950, and 3285, respectively. The number and area of hot spot streets for catering are the same as that for shopping and entertainment, but the latter contains more POIs. The number of streets for the first, second, and third-grade cold spot areas in catering is 19, 36, and 20, respectively, with areas of 1207.50 km², 2817.18 km², and 1207.50 km², and 1546, 4127, and 1546 POIs, respectively. In the same order, the three grades of cold spot areas for entertainment and shopping include 6, 22, and 26 streets, with an area of 542.05 km², 1911.81 km², and 2782.84 km², and contain 367, 5620, and 9291 POIs, respectively. Catering cold spot areas are more widely distributed and contain fewer outlets than those for shopping and entertainment; (3) the spatial distribution characteristics of accommodation hot and cold spots differ significantly from those for tourism and leisure, with all hot and cold spot streets distributed in the southeastern part of the city. The CZCF and the UFEZ are the main locations of accommodation hot spots, in addition to the three hot spots in the Miyun District of the ECZ, namely Xinchengzi Town, Gubeikou Town, and Gaoling Town. The cold spot areas are mainly distributed in Pinggu District, Shunyi District, Fangshan District, and Mentougou District. In contrast to the tourism and leisure industry, accommodation has a few hot spots in Miyun District, but no cold spots in Yanqing District (as tourism and leisure does); (4) the distribution of scenic spots is more complex. Cold spots mainly concentrate in Shunyi District and Pinggu District, while hot spot areas form three major cluster centers in UFEZ, NUDZ, and ECZ, respectively. Compared with tourism and leisure, the scenic spots cold streets

have fewer streets and smaller areas, and the CZCF is no longer a major area of scenic hot spots; (5) high-value hot spot streets for tourism and leisure, catering, accommodation, and shopping and entertainment are dominated by Beixinqiao Street, Hepingli Street, Sanlitun Street, Heping Street, and Tuanjiehu Street, indicating that there is an abundance of outlets for various types of tourism and leisure industries in and around them. The typical hot spot streets of the scenic spots are Pingguoyuan Street, Liulimiao Town, Bajiao Street, Shicheng Town, and Xiangshan Street, which do not overlap with other sectors' hot spot streets, and have relatively low Z-values and are less hot. The high-value cold spot streets of tourism and leisure, catering, accommodation, and shopping and entertainment are chiefly in Fuzizhuang Township, Wangping Town, Miaofeng Mountain Town, and Tanzhesi Town, indicating fewer various tourism and leisure industry outlets in and around them. The typical cold spot streets of the scenic spots are Beixiaoying Township, Nancai Township, Yangzhen Town, Zhangzhen Town, and Binhe Street, which do not overlap with the cold spots streets of other sectors and have a small difference in Z-values.

**Table 5.** Tourism and leisure industry Getis-Ord $G_i{}^*$ index.

| | | Tourism and Leisure | Catering | Accommodation | Scenic Spots | Shopping and Entertainment |
|---|---|---|---|---|---|---|
| **Grade I hot spot** | Street/pcs | 138 | 141 | 131 | 12 | 134 |
| | Area/km² | 1722.22 | 1788.35 | 1533.32 | 887.68 | 1583.46 |
| | POI/pcs | 249,477.00 | 76,167.00 | 20,670.00 | 291.00 | 146,009.00 |
| | Z mean value | 5.73 | 6.08 | 7.00 | 2.95 | 1.19 |
| Grade II hot spot | Street/pcs | 10 | 8 | 6 | 11 | 14 |
| | area/km² | 394.47 | 196.03 | 220.88 | 662.91 | 549.78 |
| | POI/pcs | 12,393.00 | 2534.00 | 1106.00 | 262.00 | 12,950.00 |
| | Z mean value | 2.29 | 2.24 | 2.23 | 2.28 | 2.32 |
| Grade III hot spot | Street/pcs | 5 | 5 | 3 | 13 | 5 |
| | Area/km² | 126.04 | 296.38 | 189.46 | 880.06 | 107.48 |
| | POI/pcs | 5883.00 | 2225.00 | 382.00 | 331.00 | 3285.00 |
| | Z mean value | 1.79 | 1.78 | 1.78 | 0.10 | 1.80 |
| Random distribution area | Street/pcs | 91 | 85 | 95 | 232 | 107 |
| | Area/km² | 7883.31 | 7246.53 | 10,520.84 | 59.93 | 8915.54 |
| | POI/pcs | 67,045.00 | 16,605.00 | 10,166.00 | 3553.00 | 45,749.00 |
| | Z mean value | −0.54 | −0.56 | −0.52 | 0.87 | −0.58 |
| Grade I cold spot | Street/pcs | 16 | 19 | 42 | 18 | 6 |
| | Area/km² | 1151.30 | 1207.50 | 2186.80 | 876.29 | 542.05 |
| | POI/pcs | 4823.00 | 1546.00 | 658 | 114.00 | 367.00 |
| | Z mean value | −2.76 | −2.81 | −2.91 | −2.90 | −2.80 |
| Grade II cold spot | Street/pcs | 30 | 36 | 27 | 21 | 22 |
| | Area/km² | 2371.79 | 2817.18 | 1258.18 | 891.15 | 1911.81 |
| | POI/pcs | 14,256.00 | 4127.00 | 627.00 | 155.00 | 5620.00 |
| | Z mean value | −2.18 | −2.24 | −2.56 | −2.26 | −2.24 |
| Grade III cold spot | Street/pcs | 24 | 20 | 10 | 7 | 26 |
| | Area/km² | 2743.85 | 1207.50 | 483.49 | 370.13 | 2782.84 |
| | POI/pcs | 8372.00 | 1546.00 | 252.00 | 38.00 | 9291.00 |
| | Z mean value | −1.78 | −2.81 | −1.84 | −1.79 | −1.77 |

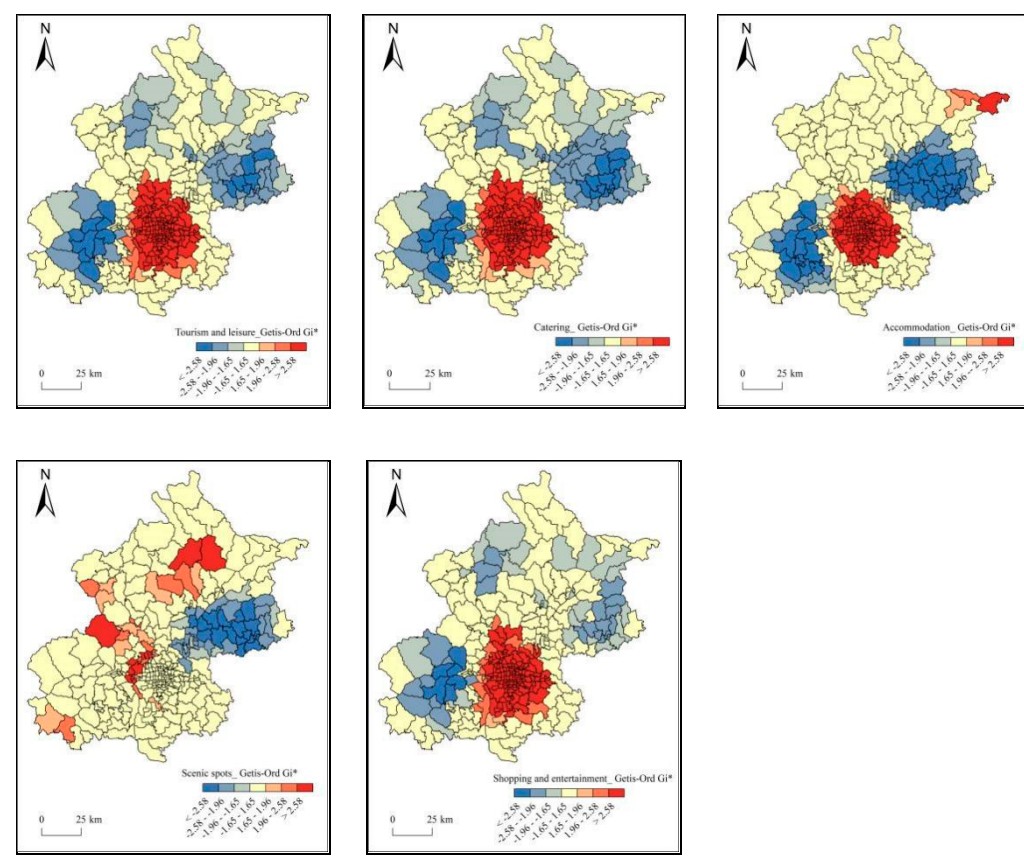

**Figure 3.** Tourism and leisure SDE and KD Getis-Ord $G_i^*$.

**Table 6.** Tourism and leisure industry typical cold and hot spot.

| Industry | | Typical Streets ($p < 0.01$) | Z Value |
|---|---|---|---|
| Tourism and Leisure | hot spot | Beixingqiao, Hepingli, Sanlitun, Hepingjie, Tuanjiehu | >7 |
| | cold spot | Fozizhuang, Wangping, Miaofeng, Tamzhesi, Heibeizhen | <−2 |
| Catering | hot spot | Beixingqiao, Tamzhesi, Sanlitun, Hepingli, Hepingjie | >8 |
| | cold spot | Fozizhuang, Dahuashan, Wangxinzhuang, Daxingzhuang, Yukou | <−2 |
| Accommodation | hot spot | Beixingqiao, Tuanjiehu, Hepingjie, Sanlitun, Hujialou | >8 |
| | cold spot | Qinglonghu, Hebei, Mulin, Tamzhesi, Beixiaoying | <−3 |
| Scenic Spots | hot spot | Pingguoyuan, Liulimiao, Bajiao, Shicheng, Xiangshan | >2 |
| | cold spot | Beixiaoying, Nancai, Yangzhen, Zhangzhen, Binhe | <−2 |
| Shopping and Entertainment | hot spot | Beixingqiao, Hepingli, Andingmen, Hepingjie, Sanlitun | >6 |
| | cold spot | Fozizhuang, Wangping, Miaofengshan, Tamzhesi, Datai | <−2 |

## 4. Discussion

### 4.1. Contribution

Previous studies on the identification of urban industry spatial structure are mostly based on conventional socio-economic statistics, census, questionnaire and interview data, and urban land use maps; research methods mainly include the rank-size rule, Pareto index, and Herfindahl index. However, due to the heavy influence of human factors, such data exhibit weaknesses both in timeliness and accuracy, and are often unable to capture the current status of urban industrial spatial structure. The open-source POI data are real-world point geographic entity data with the merits of easy access, large data volume, high accuracy, and fast updates. POI data can reveal the inner-city characteristics and reflect the spatial distribution of specific businesses in the city in more detail. To date, POI data

have been mostly used for identifying urban polycentric spatial structure, the analysis of residents' daily activity characteristics, and personalized tourism route recommendations; there are relatively few studies that have used POI data to reflect the overall structure of the urban tourism and leisure industry and its sub-sectors. In addition, this study analyzes the spatial hot spots of the tourism and leisure industry and its sub-sectors at street level, which is relatively uncommon in previous studies. Therefore, this study uses POI data and GIS spatial visualization technology to comprehensively describe the spatial layout characteristics of Beijing's tourism and leisure industry and its sub-sectors.

### 4.2. Limited Factors

Due to the lack of information on the feature of tourism and leisure industrial outlets in POI data, this study only explores the spatial distribution characteristics of outlets from the supply side. In order to show the comprehensive spatial distribution characteristics of tourism and leisure, it is necessary to combine demand data, such as outlet ratings, per capita consumption, and tourists' spatial behavior.

### 5. Conclusions

This study explores the spatial structural characteristics of Beijing's tourism and leisure industry and its sub-sectors using POI data and research instruments, including nearest neighbor index, kernel density, standard deviation ellipse, and spatial autocorrelation index. From the perspectives of spatial distribution patterns, spatial cluster characteristics, and spatial hot and cold spots, the following conclusions are drawn.

(1) In terms of the spatial distribution pattern, the tourism and leisure industry and its sub-sectors in Beijing show a clustering distribution in space, with some differences in the degree of clustering across different industries. The highest degree of clustering occurs in catering and the lowest in scenic spots;

(2) In terms of spatial cluster characteristics, detailed analysis has been done using kernel density and standard deviation ellipse after identifying the spatial distribution patterns. The tourism and leisure industry and all its sub-sectors in Beijing show a "northeast-southwest" directional trend. There are some differences in the directionality and dispersion of sub-sectors, however: the most obvious directional sub-sector is accommodation, and the most dispersive is scenic spots. Several cluster centers have been formed in Beijing's tourism and leisure industry and its sub-sectors. The degree of clustering is the highest in the CZCF and declines gradually for outer-layer districts, while high-density, contiguous clustering centers gather in the central city, and isolated island-like cluster centers are found in distant suburbs;

(3) In terms of spatial hot spot detection, the results of the global Moran's index indicate that there is a spatial cluster of high values with high values and low values with low values in the tourism and leisure industry and its sub-sectors in Beijing. Based on this and with streets as the unit of analysis, this study adopts the Getis-Ord $G_i^*$ test statistic to identify the location of hot and cold spots in specific streets in Beijing. The distribution of three grades of hot spot areas and non-significant areas of tourism and leisure, catering, accommodation, and shopping and entertainment in Beijing demonstrates a circular pattern that centers around the CZCF and expands outward in sequence. Most of the hot spot areas are distributed in the CZCF and the UFEZ, a few in the NUDZ. The distribution pattern of scenic spots is complex, with the cold spots mainly concentrated in Shunyi District and Pinggu District, but hot spots are more scattered and form three major cluster centers in the UFEZ, the NUDZ, and the ECZ, respectively. The typical hot and cold spot streets of tourism and leisure, catering, accommodation, and shopping and entertainment show a high degree of consistency, but the typical hot and cold spot streets of scenic spots do not overlap with those of the other sectors.

**Author Contributions:** M.Z. was the major writer of the manuscript; J.L. conceived the idea, led the project and acquired funding support. All authors read the first draft, helped in revision and approved the article. All authors have read and agreed to the published version of the manuscript.

**Funding:** The publication of the present work is supported by the National Natural Science Foundation of China (grant no. 41771131), Key Projects of Beijing Social Science Foundation (grant no. 21JCB050), and the Premium Funding Project for Academic Human Resources Development in Beijing Union University (grant no. BPHR2020AS02).

**Institutional Review Board Statement:** Not applicable.

**Informed Consent Statement:** Not applicable.

**Data Availability Statement:** The primary data used to support the findings of this study have been explained clearly in Section 2.2.

**Conflicts of Interest:** The authors declare that there are no conflict of interest.

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
