# Peer review of "Study on Spatial Structure Characteristics of the Tourism and Leisure Industry"

_sustainability, doi:10.3390/su132313117_

Round 1
Reviewer 1 Report
1.The research methods and writing methods of this article are complete, but there are no innovative results.
2.As discussed in landscape ecology, the development of settlements is a collection of human activities in the past few hundred years. In research information, the type of tourism analyzed is actually a presentation of a past story.
3.Conclusion There can be no new expansion and practical operation methods.
4.The practicability of the paper is low.
5.The proposal can strengthen the proposed guidelines for the public sector and private enterprises.
6.And, specify the work items and goals.
7.In addition, for the description of the national government, it is recommended to state the country's dominance of the tourism style? Is it complete guidance? Or auxiliary guidance?
Author Response
Thank you for your comments and suggestions
1.The research methods and writing methods of this article are complete, but there are no innovative results.
The authors’ Answer: In previous studies, most of the discussion on the spatial structure of the tourism industry was based on the provincial and municipal scales. This study uses POI data to explore the street-scale spatial structure of the tourism industry.
2.As discussed in landscape ecology, the development of settlements is a collection of human activities in the past few hundred years. In research information, the type of tourism analyzed is actually a presentation of a past story.
The authors’ Answer: More and more poverty-stricken areas and developing countries regard tourism as an important means of economic growth. Research on the characteristics of the spatial structure of the tourism industry is conducive to better promotion of the development of the tourism industry.
4.The practicability of the paper is low.
The authors’ Answer: The main content of this research is to explore the spatial distribution of Beijing's tourism and leisure industry.
Reviewer 2 Report
This manuscript presents a study on spatial structure characteristics of the tourism and leisure industry in Beijing. Overall, the research questions and methodology are sufficiently described. Yet, the current version suffers from two issues that need revisions before publication.
(1) The authors need to build a stronger connection between their study and sustainability, which is the core focus of the submitted journal. Undoubtedly, the tourism and leisure industry is highly relevant to sustainability and sustainable urban planning & policy, but what is the indication of its spatial structure characteristics and importance to sustainability? I would suggest the authors strengthen their Introduction section with a clearer explanation of how their research questions matter to sustainability.
(2) The current version also requires improvement with better description and presentation of the results. In section 3 (Analysis of results), the authors refer to many different districts or even specific streets in Beijing (e.g., lines 240-289). However, none of the current figures have indicated where these districts are located spatially on the maps. This lack of information confuses the readers (if they do not know much about Beijing) and difficulties in associating the results described in texts and data visualization in figures. Thus, the authors need to figure out how to indicate the significant spatial components (e.g., districts) graphically on the maps to better present the results.
Once the authors address the above issues in their revised version, this study would be appropriate to be published. The findings will contribute to a better understanding of spatial patterns of tourism and leisure spots and their relevance to sustainability.
Author Response
Thank you for your comments and suggestions
1.The authors need to build a stronger connection between their study and sustainability, which is the core focus of the submitted journal. Undoubtedly, the tourism and leisure industry is highly relevant to sustainability and sustainable urban planning & policy, but what is the indication of its spatial structure characteristics and importance to sustainability? I would suggest the authors strengthen their Introduction section with a clearer explanation of how their research questions matter to sustainability.
The authors’ Answer: We have added some explanation about the connection between tourism and sustainablity.
- The current version also requires improvement with better description and presentation of the results. In section 3 (Analysis of results), the authors refer to many different districts or even specific streets in Beijing (e.g., lines 240-289). However, none of the current figures have indicated where these districts are located spatially on the maps. This lack of information confuses the readers (if they do not know much about Beijing) and difficulties in associating the results described in texts and data visualization in figures. Thus, the authors need to figure out how to indicate the significant spatial components (e.g., districts) graphically on the maps to better present the results.
The authors’ Answer: Because of the limitation of map, we can’t indicate these districts on the map.
Reviewer 3 Report
I think this is a good job. I just have some small comments.
1. On the section of research methodology, the references are very limited. Please add more references for each method.
2. Please add more description about the situation of the tourism and leisure industry in the study area.
Author Response
Thank you for your comments and suggestions
I think this is a good job. I just have some minor comments.
- On the section of research methodology, the references are very limited. Please add more references for each method.
The authors’ Answer:We have supplemented the references and marked it in the manuscript.
- Please add more description about the situation of the tourism and leisure industry in the study area.
The authors’ Answer:We have added the situation of the tourism and leisure industry in the study area and marked it in the manuscript.
Reviewer 4 Report
The paper is a descriptive study of the spatial distribution of tourist related POIs in Beijing area. Several metrics that capture spatial trends are computed and mapped. The use of POIs as data source is promoted as an alternative to paper surveys. Some remarks below:
1) There are a few paragraphs that need proofreading, for instance at lines 46,55,170. In some sentences, clauses suggest causation but the statements end up being a bit confusing.
2) The description of factors in several metric's equations are missing or confusing. For instance in equation 1, re and Si are not explained. Also, It seems strange that R is a ratio but the ratio is the result of a summation that includes the numerator.
3) To make the findings more compelling, evidence of the advantage of using POIs over paper-survey data base maps should be provided or even better a stronger link with sustainability issues. Otherwise, the results are merely descriptive of the current situation.
4) I understood that data is not available for other researcher.
Author Response
Thank you for your comments and suggestions
The paper is a descriptive study of the spatial distribution of tourist related POIs in Beijing area. Several metrics that capture spatial trends are computed and mapped. The use of POIs as data source is promoted as an alternative to paper surveys. Some remarks below:
1.There are a few paragraphs that need proofreading, for instance at lines 46,55,170. In some sentences, clauses suggest causation but the statements end up being a bit confusing.
The authors’ Answer:We have adjusted the statements and marked them in the manuscript.
2.The description of factors in several metric's equations are missing or confusing. For instance in equation 1, re and Si are not explained. Also, It seems strange that R is a ratio but the ratio is the result of a summation that includes the numerator.
The authors’ Answer: Ratio don’t merely mean the result of a summation that includes the numerator. We have already explain the specific meanings of the factors in several metric's equations and marked them in the manuscript.
3.To make the findings more compelling, evidence of the advantage of using POIs over paper-survey data base maps should be provided or even better a stronger link with sustainability issues. Otherwise, the results are merely descriptive of the current situation.
The authors’ Answer:In the process of collecting the questionnaire, the data may be affected by the human error of the investigator,POI data is more objective and accurate.
Round 2
Reviewer 1 Report
--